# Pharmacological Immunomodulation via Collagen–Polyvinylpyrrolidone or Pirfenidone Plays a Role in the Recovery of Patients with Severe COVID-19 Through Similar Mechanisms of Action Involving the JAK/STAT Signalling Pathway: A Pilot Study

**DOI:** 10.3390/arm93040024

**Published:** 2025-07-18

**Authors:** Hugo Mendieta-Zerón, Esteban Cruz-Arenas, Salvador Díaz-Meza, Alejandro Cabrera-Wrooman, Edna Ayerim Mandujano-Tinoco, Rosa M. Salgado, Hugo Tovar, Daniel Muñiz-García, Laura Julieta Orozco-Castañeda, Sonia Hernández-Enríquez, Miriam Deyanira Rodríguez-Piña, Ana Sarahí Mulia-Soto, José Meneses-Calderón, Paul Mondragón-Terán, Edgar Krötzsch

**Affiliations:** 1Facultad de Medicina, Universidad Autónoma del Estado de México, Toluca 50000, State of Mexico, Mexico; miriam_drp@hotmail.com (M.D.R.-P.); muliasotoana205@gmail.com (A.S.M.-S.); uci20042000@yahoo.com.mx (J.M.-C.); 2Hospital Materno-Perinatal ‘Mónica Pretelini Sáenz’, Toluca 50140, State of Mexico, Mexico; 3Epidemiological Vigilance Unit, Instituto Nacional de Rehabilitación ‘Luis Guillermo Ibarra Ibarra’, Mexico City 14389, Mexico; drest.cruz.inv@gmail.com; 4Hospital General ‘Dr. Nicolás San Juan’, Toluca 50010, State of Mexico, Mexico; sdiazm@uaemex.mx (S.D.-M.); daniel_mg94@outlook.es (D.M.-G.); laurajulietaorozco@yahoo.com.mx (L.J.O.-C.); boyfriendnavi_5nph@hotmail.com (S.H.-E.); 5Laboratory of Connective Tissue, Centro Nacional de Investigación y Atención de Quemados, Instituto Nacional de Rehabilitación ‘Luis Guillermo Ibarra Ibarra’, Mexico City 14389, Mexico; aca_w@yahoo.com.mx (A.C.-W.); joaned17@hotmail.com (E.A.M.-T.); salgado_rm@yahoo.com.mx (R.M.S.); 6Computational Genomics Division, National Institute of Genomic Medicine, Mexico City 14610, Mexico; hatovar@inmegen.gob.mx; 7CICATA Unidad Morelos, Instituto Politécnico Nacional, Xochitepec 62790, Morelos, Mexico; p.mondragonteran@gmail.com

**Keywords:** collagen–polyvinylpyrrolidone, pirfenidone, inflammation, SARS-CoV-2

## Abstract

**Highlights:**

**What are the main findings?**
Collagen–polyvinylpyrrolidone and pirfenidone have been shown to be therapeutically useful in patients with severe COVID-19.Collagen–polyvinylpyrrolidone or pirfenidone could control the inflammation associated with COVID-19, perhaps by regulating the JAK/STAT pathway.

**What is the implication of the main findings?**
Collagen–polyvinylpyrrolidone and pirfenidone can be considered as adjuvant therapies for the treatment of infectious diseases involving exacerbated inflammation.

**Abstract:**

The therapeutic target of COVID-19 is focused on controlling inflammation and preventing fibrosis. Collagen–polyvinylpyrrolidone (collagen-PVP) and pirfenidone both have the ability to control the cytokine storm observed in rheumatic and fibrotic disorders. In this work, our aim was to understand the benefits of treatment with each of these drugs in patients with severe COVID-19. In total, 36 patients were treated with dexamethasone and enoxaparin, but 26 were allocated collagen-PVP or pirfenidone (*n* = 15 and 11, respectively); the clinical and metabolic effects were compared among them. Since pirfenidone works via transcriptional mechanisms, we performed a human genome microarray assay using RNA isolated from fibroblast and monocyte cultures treated with the biodrug, with the aim of hypothesising a possible mechanism of action for collagen-PVP. Our results showed that hospital stay duration, quick COVID-19 severity index (qCSI), and admission to the intensive care unit were statistically significantly lower (*p* < 0.02) in patients treated with collagen-PVP or pirfenidone when compared with the control group, and that only collagen-PVP normalised serum glucose at discharge. Ingenuity Pathway Analysis showed that the cell cycle, inflammation, and cell surface–extracellular matrix interactions could be regulated with collagen-PVP via the downmodulation of proinflammatory cytokines, while Th2 anti-inflammatory response signalling could be upregulated. Furthermore, the downregulation of some of the genes involved in nitric oxide production showed a possible control for JAK in the IFN-γ pathway, allowing for the possibility of controlling inflammation through the JAK/STAT pathway, as has been observed for pirfenidone and other immunomodulators, such as ruxolitinib.

## 1. Introduction

The coronavirus disease of 2019 (COVID-19) is a life-threatening disease and has challenged even the best health systems in the world [1,2]. Cytokine storm is considered the main cause of severity and death in patients with COVID-19, while progressive pulmonary fibrosis, derived from acute respiratory distress syndrome (ARDS), has a worse prognosis; both are possible therapeutic targets [3]. The symptoms of COVID-19 are consistent with an acute inflammatory reaction and pneumonia, which progresses to ARDS in a subset of patients [4]. Progressive dyspnoea and hypoxia are the main reasons for hospitalisation [5]; 30% of patients require admission to the intensive care unit (ICU), and 10–17% undergo intubation [4]. In laboratory studies, neutrophilia and lymphopenia were correlated with the severity of the disease [5,6]. The cytokine profile in patients with COVID-19 reveals significant increases in proinflammatory cytokines that correlate with severity [4,7]. Cytokine storm is a life-threatening immune syndrome characterised by the elevated activation of immune cells and circulating proinflammatory cytokines, lymphopenia, thrombosis, and the massive infiltration of mononuclear cells into multiple organs [8]; it is considered the leading cause of disease severity and death in patients with COVID-19 [3]. Patients with cytokine storm experience a failure of negative feedback mechanisms that prevent hyperinflammation and the overproduction of inflammatory cytokines, leading to respiratory dysfunction. During infection with the severe acute respiratory syndrome coronavirus 2 (SARS-CoV-2), a wide variability in human responses has been observed; in mild and asymptomatic patients with COVID-19, a slight delay in innate and adaptive immune responses has been determined, the cause of which has been associated with low levels of type I and type III interferons (IFNs); however, this effect is magnified in severe COVID-19, wherein the lack of IFN allows virus replication, and the prime of the immune response is delayed; however, in the following phase, the levels of proinflammatory cytokines increase above the normal innate response [9]. Similarly, with low levels of IFN, epithelial cells, macrophages, and dendritic cells express high levels of proinflammatory cytokines [10] that promote the priming of the lung endothelium by cytokine receptors that activate signalling pathways related to nuclear factor kappa-light-chain-enhancer of activated B cells (NF-kB), mitogen-activated protein kinase (MAPK), signal transducer and activator of transcription (STAT) 3, and myeloid differentiation primary response (MyD) 88; therefore, endothelial cells overexpress adhesion molecules, such as selectins and adhesion proteins, that facilitate leukocyte recruitment [11] and trigger cytokine storm. Specifically, during SARS-CoV-2 infection, interleukin (IL)-1β, IL-6, IL-8, IL-10, IL-12, IP-10, monocyte chemoattractant protein (MCP)-1, tumor necrosis factor (TNF)-α, and IFN-γ are elevated, although their levels vary according to the genetic characteristics of the patient and underlying comorbidities [12]. Furthermore, among COVID-19 patients who develop ARDS and survive to the acute phase of the disease, a substantial proportion die as a result of progressive pulmonary fibrosis, resulting from inflammation and the long-term deterioration of lung function. The pathogenesis of pulmonary fibrosis includes the deregulated release of matrix metalloproteinases during the inflammatory phase, resulting in epithelial and endothelial injury with uncontrolled fibroproliferation [13,14]. Based on experience with other diseases characterised by cytokine storm, early intervention is essential to avoid life-threatening tissue damage. In this regard, different treatment options have been considered to limit inflammatory overactivation, such as corticosteroids, monoclonal antibodies directed against certain cytokines and/or their receptors, antiviral drugs, vaccines, oligonucleotides, peptides, and interferons [15]. Corticosteroids have been used primarily for the treatment of COVID-19. Treatment with methylprednisolone reduced the risk of death (HR, 0.38; 95% CI, 0.20–0.72) among patients with ARDS, suggesting its possible intervention in cytokine storm [16]. Although corticosteroids should not be the main focus, due to the susceptibility of the patient to infections, their administration decreased the rate of admission to the intensive care unit; treated patients showed a lower need for mechanical ventilation, shorter hospital stays, and better survival rates [17]. Desperate efforts to control cytokine storm in patients with severe COVID-19 have led to the use of specific immunomodulators, such as anakinra [18] and rilonacept [19], in order to regulate the expression of IL-1, as well as monoclonal antibodies against IL-6, such as tocilizumab [20] and sarilumab [21], and IFNβ-1 [22]. Tocilizumab [23], alone or in combination with anakinra and immunoglobulin, or the addition of hydroxychloroquine and chloroquine, mesenchymal stem cells, and plasma from convalescent patients, have shown relatively positive results due to reductions in ferritin, C-reactive protein, and D-dimer, together with improvements in vascular and respiratory signs and symptoms [24]. Additionally, antifibrotic therapies could be valuable in preventing severe COVID-19 in patients with idiopathic pulmonary fibrosis and could play a role in the prevention of fibrosis after SARS-CoV-2 infection [25]. In this work, we considered two strategies for the adjuvant treatment of patients with severe COVID-19. First, we used collagen–polyvinylpyrrolidone (collagen-PVP), which is a copolymer prepared through the gamma irradiation of pepsinated porcine type I collagen and low-molecular-weight polyvinylpyrrolidone [26], since this biopharmaceutical can downmodulate overexpressed proinflammatory cytokines in hypertrophic scarring [27] and scleroderma [28], among other conditions, prevent the fibrotic component derived from chronic inflammation during cyclosporine nephrotoxicity [29], and be used in the treatment of peritoneal adhesions [30]. On the other hand, proinflammatory cytokines can be inhibited by administering pirfenidone, or 5-methyl-1-phenyl-2-(1H)-pyridone, which belongs to the pyridone class [31]; hence, it has been used in the treatment of fibrotic diseases, particularly for the treatment of idiopathic pulmonary fibrosis [32], as well as hepatic fibrosis [33], among others; in this sense, collagen-PVP and pirfenidone can be considered for the treatment of severe COVID-19 because both drugs can decrease cytokine storm during the early phase of SARS-CoV-2 infection [34,35]. Furthermore, even though the virus is eradicated in patients who have recovered from COVID-19, eliminating the cause of lung damage does not in itself prevent the development of progressive irreversible fibrotic interstitial lung disease. With this trial, we could offer alternative information to prevent pulmonary fibrosis in patients who have suffered severe COVID-19 with an immunomodulant treatment based on collagen-PVP or pirfenidone, representing a turning point in the treatment of intubated patients to increase the probability of recovery.

## 2. Materials and Methods

### 2.1. Clinical Follow-Up

A prospective and longitudinal clinical trial was conducted with patients infected with SARS-CoV-2 between February 2020 and June 2021 at the General Hospital “Dr. Nicolás San Juan”, Health Institute of the State of Mexico (ISEM), Toluca, Mexico. The patient sampling method was performed using classic continuous sequential allocation. All patients were adults, hospitalised, and had COVID-19 confirmed using polymerase chain reaction (PCR) testing, with total bilirubin ≤ 1.5 and alanine transaminase (ALT) > 5 times the upper limits relative to normal values. Patients were excluded if they were treated with biological antirheumatic drugs, disease modifiers (DMARD), or other immunosuppressive agents; required continuous therapy with systemic corticosteroids at a dose greater than 10 mg of prednisone per day or equivalent; were pregnant women, nursing or intending to become pregnant during the study; required continuous treatment with strong cytochrome P450 (CYP)1A2 inhibitors (that is, Fluvoxamine, Enoxacin); had a calculated creatinine clearance (or estimated glomerular filtration rate) < 10 mL/min; or required renal replacement therapy.

For the anthropometric evaluation of each patient, the weight record and blood pressure of the participants were evaluated electronically in bed; height was measured with a conventional stadiometer, and body mass index (BMI) was calculated. Body temperature, heart rate, respiratory rate, stands for saturation of peripheral oxygen (SpO2), the partial pressure of carbon dioxide (PCO2), and the partial pressure of oxygen (PO2) were monitored during the hospitalisation of each patient. The Kirby index, the quick COVID-19 severity index (qCSI), and the shock index were evaluated at admission and discharge. Haematic biometry analyses, including serum concentrations of glucose, uric acid, cholesterol and triacyl glycerides, blood gas, blood urea nitrogen, and creatinine, were routinely performed from blood samples obtained in the fasting state according to the International Federation of Clinical Chemistry and Medical Laboratories (IFCC). Serum samples were obtained at the beginning and after 7 days of treatment and stored at −80 °C for a further evaluation of representative levels of Th1, Th2, and Th17 cytokines (IFN-γ, TNF-α, IL-2, -4, -10, -13, and -17), according to the manufacturer (R&D Systems, Inc., Minneapolis, MN, USA).

Upon acceptance with the signing of informed consent, patients received one of the following treatments: pirfenidone (KitosCell tabs, CellPharma S de RL de CV., Mexico City, Mexico), 1200 mg of oral q12 h; or collagen-PVP (Fibroquel, Aspid SA de CV, Mexico City, Mexico), 2 mL of intramuscular q24 h; both for 7 days; the control group comprised patients without pirfenidone or collagen-PVP.

### 2.2. Ethics

This trial was accepted by the Research Ethics Committee of the “Mónica Pretelini Sáenz” Maternal-Perinatal Hospital (HMPMPS) (code 2020-12-712), with current registration in the National Bioethics Commission (CONBIOETICA), as well as by the Research Committee of the same Hospital, with registration in the Comisión Federal para la Protección contra Riesgos Sanitarios (COFEPRIS). Furthermore, the trial was registered on ClinicalTrial.gov with the ID: NCT06585319. This research was carried out under the deontological considerations recognised by the Declaration of Helsinki (Fortaleza, Brazil, 2013), and, according to the level of intervention, it is considered a study with greater than minimal risk.

### 2.3. Human Genome Microarray Assay for Fibroblasts and Monocytes Treated In Vitro with Collagen-PVP

The human telomerase reverse transcriptase (hTERT)-immortalised fibroblasts-BJ1 were cultured with Dulbecco’s modified Eagle’s Medium (Gibco, Life Technologies, Grand Island, NY, USA), supplemented with 10% foetal bovine serum (FBS, Gibco), 2 mM L-glutamine (Gibco), 100 U/mL penicillin, and 100 μg/mL streptomycin (Gibco). The human monocyte cell line (THP-1) was cultured with Roswell Park Memorial Institute (RPMI)-1640 medium (Gibco), supplemented with 10% FBS, 1 mM of sodium pyruvate, 0.1 mM of non-essential amino acids, 0.1 mM of glutamine, 100 U/mL penicillin, and 100 μg/mL streptomycin (Gibco). Cultures were kept at 37 °C in a 5% CO_2_ atmosphere. Adherent THP-1 cells were serially subcultured until the appropriate cell concentration was reached to perform the experiment. Despite our experience of in vitro cell tolerance to different concentrations of collagen-PVP (1–10%) [26,36], prior to the genome microarray assay, the viability of the fibroblast and monocyte cultures was evaluated using the incorporation of thymidine. For collagen-PVP treatments, 2 × 10^5^ cells, fibroblasts, and monocytes were seeded in 5 mL of their corresponding medium in T-25 culture flasks (Corning, Glendale, AZ, USA) and kept for 24 h at 37 °C and 5% CO_2_. Cultures were incubated for 3 h with FBS-free medium, and then with 3% collagen-PVP prepared in the corresponding medium without FBS, and incubated for 24 h at 37 °C and 5% CO_2_. The medium was removed and the flasks were washed with phosphate saline buffer. Total RNA was isolated from collagen-PVP-treated and control cells using the TRIzol method (Invitrogen™. Thermo Fisher Scientific Inc., Waltham, MA, USA). RNA expression was analysed using the Genechip Human Mapping 10 K v1.0 microarray, developed at the Core Microarray Facility at the Instituto de Fisiología Celular, UNAM. GenArise 1.84.0 software was used for the quantification of microarray data with a cut-off value ≥ 2.0 for upregulation and ≤−2.0 for downregulation. The target genes were analysed using DAVID Bioinformatics Resources 6.8 (NAID, NIH, Bethesda, MD, USA). For the analysis of biological function and signalling regulation, an enrichment analysis of the up- and downregulated genes was performed using the Ingenuity Pathway Analysis (IPA, Ingenuity Systems Inc., Redwood City, CA, USA). General Z-score values and raw data can be downloaded with the following gene expression omnibus (GEO) accession numbers: GSE262736 and GSE262737. IPA generates copy numbers (CN) relying on a highly curated knowledge-based source, the Ingenuity Knowledge Base (IKB), and the IKB reports a series of experimentally observed cause–effect relationships related to transcription, expression, activation, molecular modification, binding events, and transport processes. Since these interactions have been experimentally measured, they can be associated with the definite direction of the causal effect, and either the activation or inhibition of the processes mentioned above at a whole-genome network-wide level. Further methodological details of the CN analysis that we performed are described in [37].

### 2.4. Statistical Analysis

From clinical and biochemical data, frequency measures (percentage) were initially obtained for categorical variables, while median values ± standard deviation (SD) were obtained for continuous variables. The stratified variables were then compared according to the type of treatment using the chi-square test (χ2) and analysis of variance (ANOVA) for categorical variables, while the Kruskal–Wallis test was performed for continuous variables. The normal distribution was verified using the Shapiro–Wilk test. Statistically significantly different values were those with *p* values ≤ 0.05. Stata v13.0 (StataCorp., College Station, TX, USA) was used to perform all statistical analyses.

## 3. Results

### 3.1. Treatment of Patients with COVID-19 with Collagen-PVP or Pirfenidone Improves Some Disease Indicators

Thirty-six patients, adults with an average age of 50.8 years old (range 19–84), were enrolled in this work, and all maintained the same base treatment (dexamethasone and enoxaparin) during their hospitalisation period. Groups of 15 and 11 patients received one of the two immunomodulators evaluated, collagen-PVP or pirfenidone, respectively (Figure 1).

Since all patients were received by the hospital under severe infection conditions, 41% of them died during the trial due to complications from COVID-19, and no statistically significant differences were associated with the treatment group (Table 1).

Although there was no difference between groups in the reason for discharge, the mean follow-up stay (inpatient stay) duration indeed exhibited statistically significant differences when patients were treated with immunomodulators: collagen-PVP and pirfenidone for 12.7 and 13.2 days, respectively, vs. 24.1 days for the control group (*p* = 0.015, Table 2). The discharge qCSI and serum glucose indicated statistically significant differences when the three groups were compared (*p* < 0.02 for both indicators, Table 2). In all cases, the values increased from collagen-PVP to controls, where no statically significant differences were observed between the immunomodulating treatments. In particular, at discharge, glucose values were normal only in the group of patients treated with collagen-PVP.

Finally, it was more common for patients in the control group to be admitted to the intensive care unit than for those treated with immunomodulators (*p* = 0.012), and, again, in the same proportion as previously mentioned, the lowest number of ICU admissions was for patients who underwent collagen-PVP treatment (Table 2). Regarding the evaluation of serum cytokines, blood values were below the detection levels of the kits in the majority of patients; IFN-γ (5.7 pg/mL), TNF-α (6.2 pg/mL), IL-2 (7 pg/mL), IL-4 (10 pg/mL), IL-10 (3.9 pg/mL), IL-13 (57.4 pg/mL), and IL-17 (15 pg/mL); in such a biochemical analysis, the values of cytokines involved in systemic inflammation were not available.

### 3.2. Collagen-PVP Modulates Different Signalling Pathways According to Cell Lineage

Since immunomodulators, including pirfenidone [38], have demonstrated the downregulation of inflammation during COVID-19 treatment [39], we considered performing a presumptive analysis of the effects of collagen-PVP on inflammatory gene expression under basal (unstimulated) conditions in stromal and immune cells. Both cell types showed interesting changes in gene expression after collagen-PVP treatment, although fibroblasts showed the biggest changes (twice as up- or downregulated as monocytes, as shown in Table 3).

In particular, monocytes treated with collagen-PVP increased the number of genes related to inflammation that involved the Th2 response and the signalling of cell extracellular matrix receptors (Table 4).

On the other hand, fibroblasts negatively regulated the cell cycle, inflammation, and cell extracellular matrix receptor signalling (Figure A1 and Figure A2 and Table 4), while monocytes decreased those related to IL-6 signalling (Figure A3 and Table 4). Furthermore, the shared signalling pathways between fibroblasts and monocytes are involved in the production of nitric oxide (Figure A4), the second messenger downregulation in extracellular matrix receptor signalling (Figure A2), and the cell cycle and differentiation (Table 5). Among the genes involved in the production of nitric oxide by inducible nitric oxide synthase, the IPA analysis showed a possible control for JAK in the pathway activated by IFN-γ through its receptor, IFN-γR, suggesting the possibility of also controlling inflammation through the JAK/STAT pathway (Figure A4).

## 4. Discussion

The need to control hyperinflammation in patients with severe COVID-19, as well as to prevent the resulting lung fibrosis after the viral crisis, has led the biomedical community to propose combined therapies; for example, the combination of remdesivir’s antiviral properties, with methylprednisolone, as an unspecific immunomodulator, demonstrated efficacy in the treatment of pneumonia associated with COVID-19. Unfortunately, bacterial infection has been associated with this treatment; therefore, in some cases, the treatment must be retired early [40], sometimes also due to side effects [41]. In the search for less aggressive treatments, but with the same objective, we and others considered the treatment of COVID-19 with collagen-PVP [34,42] or pirfenidone [26,43], in combination with dexamethasone, such that the side effects derived from cytokine storm control observed with other combinations could be avoided [41]. We observed an improvement in the pneumonic condition after treatment with collagen-PVP or pirfenidone, when compared with the controls, despite the high mortality rate still present due to the critical stage of some of the admitted patients. Since hyperglycaemia is a secondary effect resulting from the inflammation observed in patients during or after COVID-19 [44], it was striking that glucose values at the time of discharge were normal only when patients were treated with collagen-PVP, while the high values observed for the pirfenidone and control groups were maintained, indicating a possible metabolic effect derived from the control of inflammation. Another important variable measured during our study was the use of supplemental oxygen, which decreased after treatment with collagen-PVP or pirfenidone, although only at a difference close to statistical significance vs. the control group, where the sample size could have played a critical role; this was an important limitation of our work, even when data from all of the patients (*n* = 36) were considered for most outcome variables, namely in living or dying patients; for the last, the value at discharge was considered to be that obtained from the last evaluation while living.

Biochemically, high levels of circulating cytokines were associated with SARS-CoV-2 infection; COVID-19 patients expressed higher levels of Th1 cytokines, but they also showed Th2 immune profile cytokines, which biologically antagonise Th1 or the early proinflammatory response [10]. Despite several works reporting serum cytokines from patients with COVID-19 in different states of disease severity, the data are ambiguous due to the high variability of the results; some researchers reported values in units of picograms/mL [45], while others found hundreds of picograms/mL [46]. In this work, cytokine levels were undetectable in most patients, due to the sensitivity of the ELISA kits we used.

As mentioned above, pirfenidone is a drug capable of controlling inflammation [47] and pulmonary fibrosis [48]; these pharmacological properties have led investigators to consider its administration in patients with COVID-19, such that pirfenidone has been considered for early and post-COVID-19 treatments. The mechanism involved in the regulation of inflammation by pirfenidone involves the regulation of proinflammatory cytokine regulation [38] and the JAK2/STAT3 pathway [49]. Our group [27] and others [50,51] have shown that collagen-PVP exhibits fibrolytic and inflammatory control effects in different tissues and organs; however, intracellular mechanisms with respect to the effects of collagen-PVP have not yet been analysed, except for the decrease in integrin α2 vs. control, when human fibroblasts were cultured with 3% collagen-PVP [26]. Recently, collagen-PVP (named polymerised type I collagen) was found to downregulate STAT1 phosphorylation in type 1 macrophages, differentiated from the THP-1 monocyte cell line, where, apparently, the effect was driven by the binding of collagen-PVP to the leukocyte-associated immunoglobulin-like receptor 1 (LAIR1) [52]. Although these results match the IPA analysis of our work regarding JAK/STAT pathway regulation, and consequently support our hypothesis, it remains doubtful whether the target of collagen-PVP could be LAIR1; this is because, in the experiments performed by Olivares-Martínez E. et al. in 2025, a differential effect between type I collagen vs. collagen-PVP in their affinity for LAIR1 was not demonstrated [52], such that, if we consider that type I collagen and its degradation products are permanently available in different organisms, and that type I collagen and collagen-PVP are cleaved immediately in vitro by matrix metalloproteinase 1 [26], the pharmacodynamics of collagen-PVP remain unclear.

In order to better understand the pathways involved in the effects of collagen-PVP on fibroblasts and monocytes, we performed a microarray assay, an experimental approach that was designed as a first screening step to assess whether the compound could modulate the constitutive expression of inflammatory mediators in the absence of a defined inflammatory challenge, in line with previous reports on the evaluation of biomaterials, in which early assessments were conducted in resting cells to explore biocompatibility and homeostatic immunomodulation before moving to inflammatory or pathological models [53,54]. Furthermore, our objective was to explore whether certain pathways related to inflammation and fibrosis could be regulated similarly to observations of other drugs used in the treatment of COVID-19. Patients with mild COVID-19 who were treated with the intramuscular administration of collagen-PVP showed an improvement in the disease during the first week of treatment [42], where the effects were associated with reductions in IP-10, IL-8, and macrophage colony stimulating factor (M-CSF) [34]; these results are consistent with our preliminary findings, obtained through microarray assays performed in fibroblast and monocyte cultures treated with 3% collagen-PVP, where some important pathways related to inflammation, apoptosis, and ECM receptor signalling were modulated. In particular, IL-6 and IL-8 signalling was negatively regulated, as well as the JAK and MAPK signalling that would reduce the expression of iNOS favouring the anti-inflammatory condition [55] when stromal and immune cells were treated with collagen-PVP. Furthermore, we considered that a possible upregulation of the Th2 immune response pathway could play an important role in the effects observed in patients with COVID-19 treated with collagen-PVP; again, this hypothesis is also supported by the paper recently published by Olivares-Martínez E. et al. [52]. Despite the sound hypothetic mechanism reported in relation to the mentioned results, it is necessary to perform concluding experiments to demonstrate which of the genes involved in collagen-PVP treatment can regulate the pathways mentioned above. Even so, this mechanism of action, in addition to the clinical results observed after the trial, strengthens the claim that collagen-PVP can be used to treat the hyperinflammation observed in COVID-19, as well as other pathologies or conditions that share that characteristic, with the biodrug not generating any side effects related to its administration as of yet [56].

## 5. Conclusions

This is not the first time collagen-PVP and pirfenidone have been compared for the treatment of inflammation and fibrosis. Previously, both drugs demonstrated the ability to decrease inflammation, fibrosis, and the expression of pro-inflammatory/fibrogenic cytokines; here, we have shown that collagen-PVP and pirfenidone can improve COVID-19 patient recovery similarly when administered early. Furthermore, data obtained from the Ingenuity Pathway Analysis, derived from the microarray evaluation of collagen-PVP-treated fibroblasts and monocytes, suggested a mechanism of action very similar to that of pirfenidone, which suggests that they should be considered as a combined treatment in different inflammatory and/or fibrotic conditions.

## Figures and Tables

**Figure 1 arm-93-00024-f001:**
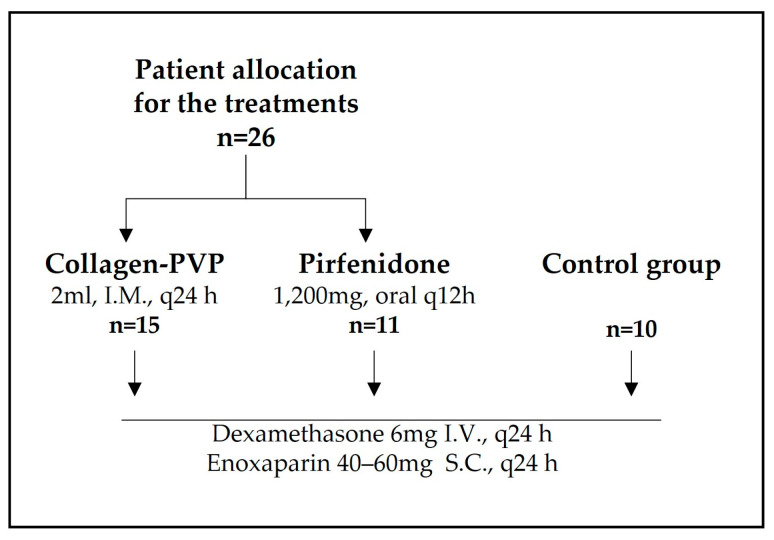
Strategic workflow and patient allocation.

**Table 1 arm-93-00024-t001:** General characteristics of the studied population.

Variable	n (%) or Mean (±SD)
Group	
A (Collagen-PVP)	15 (57.7)
B (Pirfenidone)	11 (42.3)
C (Control)	10 (27.8)
Sex	
Male	17 (47.2)
Female	19 (52.8)
Age	50.8 (19–84)
Discharge reason	
Patient improvement	21 (58.3)
Death	15 (41.7)

SD = standard deviation. Only for Age, data between parentheses indicates the range. Total *n* = 36.

**Table 2 arm-93-00024-t002:** Differences according to treatment.

Variable	Group n (%), Mean (±SD) or Median			
	Collagen-PVP *n* = 15	Pirfenidone *n* = 11	Control *n* = 10	*p* *Value*
**Sex**				0.799
Male	8 (53.3)	5 (45.5)	4 (40.0)	
Female	7 (46.7)	6 (54.5)	6 (60.0)	
**Age**	47.20 (±13.7)	56.45 (±13.0)	50.10 (±19.2)	0.406
**Discharge reason**				0.297
Patient improvement	11 (73.3)	5 (45.5)	5 (50.0)	
Death	4 (26.7)	6 (54.6)	5 (50.0)	
Follow-up days (hospital stay)	12.67 (±6.8)	13.18 (±8.1)	24.10 (±14.2)	0.015
Weight	71.47 (±13.3)	73.18 (±7.7)	68.60 (±18.1)	0.738
Size	1.62 (±0.1)	1.60 (±0.1)	1.60 (±0.1)	0.677
BMI	27.22 (±4.7)	28.90 (±4.3)	26.54 (±5.9)	0.552
Comorbidities	4 (80.0)	5 (100.0)	-	0.297
SBP-admission	120.60 (±14.4)	119.54 (±26.9)	119.80 (±15.3)	0.989
DBP-admission	71.40 (±11.9)	74.45 (±12.4)	74.40 (±17.4)	0.810
MBP-admission	87.80 (±11.7)	89.48 (±16.9)	89.53 (±16.2)	0.943
SBP-discharge	108.70 (±8.6)	104.10 (±18.4)	104.50 (±9.7)	0.717
DBP-discharge	67.00 (±7.3)	61.5 (±9.6)	67.83 (±6.4)	0.224
MBP-discharge	62.23 (±35.9)	75.70 (±12.3)	53.37 (±40.3)	0.322
HR-admission	90.13 (±15.7)	84.82 (±22.5)	84.90 (±24.6)	0.751
HR-discharge	74.30 (±16.6)	81.70 (±21.8)	88.50 (±15.0)	0.393
BF-admission	24.5	23.0	22.0	0.303
BF-discharge	21.0	23.0	18.0	0.172
Body temperature-admission	36.33 (±0.6)	36.23 (±0.8)	36.16 (±0.8)	0.846
Body temperature-discharge	36.18 (±0.4)	36.48 (±0.7)	36.67 (±0.3)	0.164
Shock index-admission	0.8	0.7	0.6	0.457
Shock index-discharge	0.7	0.7	0.9	0.390
SpO2-admission	80.0	93.0	93.5	0.059
SpO2-discharge	91.5	91.0	93.0	0.390
Use of supplemental oxygen	4 (50.0)	7 (100.0)	1 (100.0)	0.069
FiO2-admission	0.9	0.9	0.7	0.240
FiO2-discharge	0.2	0.9	0.3	0.196
Kirby-admission	96.7	90.9	130.8	0.100
Kirby-discharge	135.1	133.3	103.3	0.397
qCSI-admission	8.0	6.0	5.0	0.363
qCSI-discharge	2.0	4.5	6.0	0.019
Glucose-admission	108.5	151.5	146.0	0.397
Glucose-discharge	90.0	119.5	179.0	0.018
BUN-admission	14.5	25.5	23.0	0.568
BUN-discharge	18.0	16.0	12.0	0.673
Creatinine-admission	0.7	0.9	0.9	0.265
Creatinine-discharge	0.6	0.7	0.4	0.608
Leukocytes-admission	10.9	11.1	11.4	0.951
Leukocytes-discharge	8.1	10.6	9.3	0.110
Blood gas pH-admission	7.5	7.5	7.4	0.807
Blood gas pH-discharge	7.5	7.3	7.4	0.185
PCO2-admission	31.0	29.0	30.0	0.462
PCO2-discharge	33.0	60.5	37.0	0.219
PO2-admission	64.5	69.0	65.5	0.690
PO2-discharge	57.0	55.5	55.0	0.908
HCO3-admission	21.85 (±5.4)	19.04 (±3.7)	20.73 (±6.2)	0.462
HCO3-discharge	23.72 (±3.8)	29.38 (±8.6)	27.08 (±5.4)	0.378
Surgical interventions	2 (20.0)	5 (45.5)	-	0.217
Admission to the ICU	2 (16.7)	5 (45.5)	8 (80.0)	0.012

The text highlighted in grey shows indicators that exhibited statistically significant differences. SD = standard deviation, BMI = body mass index, SBP = systolic blood pressure, DBP = diastolic blood pressure, MBP = mean arterial pressure, HR = heart rate, BF = breath frequency, SpO2 = saturation of peripheral oxygen, FiO2 = fraction of inspired oxygen, qCSI = quick COVID-19 severity index, BUN = blood urea nitrogen, PCO2 = partial pressure of carbon dioxide, PO2 = partial pressure of oxygen, HCO3 = bicarbonate. ICU = intensive care unit.

**Table 3 arm-93-00024-t003:** Overall gene expression of fibroblasts and macrophages treated with collagen-PVP.

Fibroblasts		Monocytes	
*Up Regulated*	*Down Regulated*	*Up Regulated*	*Down Regulated*
618	620	328	140

**Table 4 arm-93-00024-t004:** Pathway signalling related to cell cycle, cell death, and inflammation possibly involved in the gene expression of fibroblasts and macrophages treated with collagen-PVP.

Fibroblasts		Monocytes	
*Upregulated*	*Downregulated*	*Upregulated*	*Downregulated*
	Cell cycle	IL-2 signalling	IL-6 signalling
	Integrin signalling	Th2 signalling	
	IL-8 signalling	Protein kinase A signalling	
	NFAT signalling on inflammation	IL-3 signalling	
		Integrin signalling	

Pathways appear in decreased order of mRNA expression as obtained from the microarray assay.

**Table 5 arm-93-00024-t005:** Pathway signalling related to cell cycle, cell death, and inflammation, possibly involved in gene expression of both fibroblast and macrophages treated with collagen-PVP.

*Upregulated*	*Downregulated*
Nitric oxide production	Apoptosis
Phospholipase C signalling	
ILK-mediated signalling	
mTOR signalling	
Gαq signalling	
Rho family GTPase- mediated signaling	
ERK/MAPK signalling	

ILK = Integrin-Linked Kinase, mTOR = Mechanistic Target of Rapamycin Kinase, Gαq = Gq protein alpha subunit, Rho = Ras homology of the Guanosine Triphosphate family proteins. ERK = Extracellular Signal-Regulated Kinases, MAPK = Mitogen-Activated protein Kinases.

## Data Availability

The trial was registered on ClinicalTrial.gov with the ID: NCT06585319. General Z-score values and raw data can be downloaded with the following GEO accession numbers: GSE262736 and GSE262737.

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
