# Peer review of "Pharmacological Immunomodulation via Collagen–Polyvinylpyrrolidone or Pirfenidone Plays a Role in the Recovery of Patients with Severe COVID-19 Through Similar Mechanisms of Action Involving the JAK/STAT Signalling Pathway: A Pilot Study"

_arm, 2025, doi:10.3390/arm93040024_

Round 1

Reviewer 1 Report

Comments and Suggestions for Authors

Many thanks to the authors for their work. Kindly consider the comments in the attached file to improve and increase the value of the work. 

Thank you, and best regards

Comments on the Quality of English Language

Only attentive editing and phrasing are needed to increase the quality of transmission and the impact on readers

Reviewer 2 Report

Comments and Suggestions for Authors

In this Article, the Authors describe and discuss the results of their pilot study concerning the "positive role played by collagen-polyvinylpyrrolidone or pirfernidone in the recovery of patients affected by severe COVID-19".

The authors of this study document the properties of the two drugs, despite, they claim that “the intracellular mechanism of action of pirrernidone is partially known”.

Collagen-polyvinylpyrrolidone is described as a drugs with well-documented anti-inflammatory and antifibrotic properties, as well as pirfernidone. (Ref. 16-19).

However, several more recent studies showed opposite results.  

Guadalupa et al. (Pirfenidone in post-COVID-19 pulmonary fibrosis: a phase 2 randomised clinical trial - June 2025) showed that “the overall improvements in lung function and HRCT fibrotic score after 6 months with pirfenidone were not significantly different than with placebo”.

The study of Barden-Garcia (Effects of anti-fibrotic standard of care drugs on senescent human lung fibroblasts. 2022) claims that the”treatment with pirfenidone significantly increases the ratio p-MLKL/MLKL, concluding that this drug leads to necroptosis (which is in contrast with anti-inflammatory properties).

At light of this, the scientific basis appears to be rather weak.

In addition, the clinical follow up described in par. 2.1, is no significant, since the limited number of enrolled patients, together with the relevant number of deaths (41%) which did not allow to calculate statistically significant differences between the groups.

Biochemical analysis of cytokines involved in inflammatory response associated with Sars-Cov 2 infection has not been carried out.

At the same way, in par. 2.3, the downregulation of inflammatory response after cell treatment with Collagen-polyvinylpyrrolidone cannot be considered valid, because the cells have not been previously activated by an inflammatory stimulus, such as a viral infection.

It would have been more suggestive to assess the gene expression of the cell cultures in absence of an inflammatory stimulus, after the stimulation itself and after addition of the Collagen-polyvinylpyrrolidone.

The article also contains a number of formal errors (e.g. pag 2: Furthermore, in COVID-19, many of the patients……; pag. 3. Par. 2.1: Patients were excluded i treated….)

Comments on the Quality of English Language

The article also contains a number of formal errors (e.g. pag 2: Furthermore, in COVID-19, many of the patients……; pag. 3. Par. 2.1: Patients were excluded i treated….)

Reviewer 3 Report

Comments and Suggestions for Authors

In this study, the author studied the Pharmacological immunomodulation of collagen-polyvinylpyrrolidone or pirfenidone in the recovery of patients with severe COVID-19. The manuscript has the following shortcomings that need to be improved.

  1. Only 36 patients were enrolled in this study, which is a relatively small number.
  2. This study lacks necessary in vivo or in vitro validation experiments.

For the issue of a small number of research cases, it is recommended that the author provide an explanation and supplement it in the limitations section of the manuscript.

For the missing validation experiments, it is recommended that the author supplement some validation experiments to better support the conclusions of the manuscript.

Round 2

Reviewer 2 Report

Comments and Suggestions for Authors

At the light of the new revisions, I believe that your study can be published, provided that it is always a pilot study

Author Response

Dear Reviewer: Thank you very much for your advice. We acknowledge your review has improved our work.

Reviewer 3 Report

Comments and Suggestions for Authors

None

Author Response

(The authors gave the same response as above.)
